# Possible intermediate quantum spin liquid phase in $\alpha$-RuCl$_3$ under high magnetic fields up to 100 T

Xu-Guang Zhou ®[1,6], Han Li[2,3,6], Yasuhiro H. Matsuda ®[1] ✉, Akira Matsuo[1], Wei Li ®[3,4] ✉, Nobuyuki Kurita ®[5], Gang Su ®[2], Koichi Kindo[1] & Hidekazu Tanaka[5]

Pursuing the exotic quantum spin liquid (QSL) state in the Kitaev material $\alpha$-RuCl$_3$ has intrigued great research interest recently. A fascinating question is on the possible existence of a field-induced QSL phase in this compound. Here we perform high-field magnetization measurements of $\alpha$-RuCl$_3$ up to 102 T employing the non-destructive and destructive pulsed magnets. Under the out-of-plane field along the $\mathbf{c}^*$ axis (i.e., perpendicular to the honeycomb plane), two quantum phase transitions are uncovered at respectively 35 T and about 83 T, between which lies an intermediate phase as the predicted QSL. This is in sharp contrast to the case with in-plane fields, where a single transition is found at around 7 T and the intermediate QSL phase is absent instead. By measuring the magnetization data with fields tilted from the $\mathbf{c}^*$ axis up to 90° (i.e., in-plane direction), we obtain the field-angle phase diagram that contains the zigzag, paramagnetic, and QSL phases. Based on the $K$-$J$-$\Gamma$-$\Gamma'$ model for $\alpha$-RuCl$_3$ with a large Kitaev term we perform density matrix renormalization group simulations and reproduce the quantum phase diagram in excellent agreement with experiments.

Quantum spin liquid (QSL) constitutes a topological state of matter in frustrated magnets, where the constituent spins remain disordered even down to absolute zero temperature and share long-range quantum entanglement[1–4]. Due to the lack of rigorous QSL ground states, such ultra quantum spin states are less well-understood in systems in more than one spatial dimension before Alexei Kitaev introduced the renowned honeycomb model with bond-dependent exchange[5]. The ground state of the Kitaev honeycomb model is proven to be a QSL with two types of fractional excitations[5,6]. Soon after, the Kitaev model was proposed to be materialized in the $J_{\mathrm{eff}} = 1/2$ Mott insulating magnets[7–11] such as A$_2$IrO$_3$ (A = Li and Na)[12,13], $\alpha$-RuCl$_3$[14,15], etc.

Recently, the 4$d$ spin-orbit magnet $\alpha$-RuCl$_3$ has been widely accepted as a prime candidate for Kitaev material[16–22]. As initially proposed from the first-principle analysis[14,15,23–25], the compound is now believed to be described by the $K$-$J$-$\Gamma$-$\Gamma'$ effective model that includes the Heisenberg $J_{(1,3)}$, Kitaev exchange $K$, and the symmetric off-diagonal exchange $\Gamma^{(')}$ terms. The Kitaev interaction originates from chlorine-mediated exchange through edge-shared octahedra arranged on a honeycomb lattice. Similar to the intensively studied honeycomb and hyperhoneycomb iridates[26], additional non-Kitaev terms $\Gamma^{(')}$ and/or $J_3$, unfortunately, stabilize a zigzag antiferromagnetic order below $T_N \approx 7$ K in the compound[17,18,20,27]. Given that, a natural approach to realizing the Kitaev QSL is to suppress the zigzag order by applying magnetic fields to the compound[28–42]. As shown in certain experiments, a moderate in-plane field (about 7 T) can suppress the zigzag order and may induce an intermediate QSL phase before the

[1]Institute for Solid State Physics, University of Tokyo, Kashiwa, Chiba 277-8581, Japan. [2]Kavli Institute for Theoretical Sciences, University of Chinese Academy of Sciences, 100190 Beijing, China. [3]Peng Huanwu Collaborative Center for Research and Education & School of Physics, Beihang University, 100191 Beijing, China. [4]CAS Key Laboratory of Theoretical Physics, Institute of Theoretical Physics, Chinese Academy of Sciences, 100190 Beijing, China. [5]Department of Physics, Tokyo Institute of Technology, Tokyo 152-8551, Japan. [6]These authors contributed equally: Xu-Guang Zhou, Han Li. ✉e-mail: ymatsuda@issp.u-tokyo.ac.jp; w.li@itp.ac.cn

polarized phase[34,35,39–41]. However, there are also experimental pieces of evidence from, e.g., magnetization[18,27], magnetocaloric[43], magneto-torque measurements[44], etc., that indicate a single transition scenario with no intermediate phase present. Some angle-dependent experiments, on the other hand, demonstrate the presence of an additional intermediate phase, which however is, due to another zigzag anti-ferromagnetic order induced by six-layer periodicity along the out-of-plane direction[45]. This leaves an intriguing question to be resolved in the compound $\alpha$-RuCl$_3$.

Theoretical progress lately suggests the absence of intermediate QSL under in-plane fields, while predicting the presence of an inter-mediate phase by switching the magnetic fields from in-plane to the much less explored out-of-plane direction. The numerical calculations[46–50] of the $K$-$J$-$\Gamma$-$\Gamma'$ spin model show that the off-diagonal exchanges $\Gamma^{(\prime)}$ terms dominate the magnetic anisotropy in the com-pound. Due to the strong magnetic anisotropy in $\alpha$-RuCl$_3$, the critical field increases dramatically from the in-plane to the out-of-plane direction. The authors in ref. 47 further point out an interesting two-transition scenario with a field-induced intermediate QSL phase, which is later confirmed by other theoretical calculations[49], except for sub-tlety in lattice rotational symmetry breaking (such a so-called nematic order is, however, not directly relevant to our experimental discussion here as the realistic compound $\alpha$-RuCl$_3$ does not strictly have a $C_3$ symmetry[15,23,30]). More recently, H. Li et al. proposed a large Kitaev-term spin Hamiltonian[51] also based on the $K$-$J$-$\Gamma$-$\Gamma'$ model. With the precise model parameters determined from fitting the experimental thermodynamics data, they theoretically reproduced the suppression of zigzag order under the 7-T in-plane field, and find a gapless QSL phase located between two out-of-plane transition fields that are about 35 T and of 100-T class, respectively. Therefore, the previously unsettled debates on the field-induced transitions and the concrete theoretical proposal of the intermediate QSL phase strongly motivate a high-field experimental investigation on $\alpha$-RuCl$_3$ along the out-of-plane direction and up to 100 T.

In this work, we report the magnetization ($M$) process of $\alpha$-RuCl$_3$ by applying magnetic fields ($H$) in various directions within the hon-eycomb plane and along the $\mathbf{c}^*$ axis (out-of-plane) up to 100 T, and find clear experimental evidence supporting the two-transition scenario. Here, the $\mathbf{c}^*$ axis is the axis perpendicular to the honeycomb plane[27]. Under fields applied along and close to the $\mathbf{c}^*$ axis, an intermediate phase is found bounded by two transition fields $H_c^l$ and $H_c^h$. In parti-cular, besides the previously reported $H_c^l \simeq 32.5$ T[44,52], remarkably we find a second phase transition at a higher field $H_c^h \simeq 83$ T. Below $H_c^h$ and above $H_c^l$ there exists an intermediate phase − the predicted field-induced QSL phase[47,51]. When the field tilts an angle from the $\mathbf{c}^*$ axis by 9°, only the transition field $H_c$ is observed, indicating the intermediate QSL phase disappears. Accordingly, we also perform the density-matrix renormalization group (DMRG) calculations based on the pre-viously proposed $K$-$J$-$\Gamma$-$\Gamma'$ model of $\alpha$-RuCl$_3$, and find the simulated phase transitions and extended QSL phase are in agreement with experiments. Therefore, we propose a complete field-angle phase diagram and provide the experimental evidence for the field-induced QSL phase in the prominent Kitaev compound $\alpha$-RuCl$_3$.

## Results
### Experimental results
Figure 1a−c shows the magnetization process and the magnetic field dependence of d$M$/d$H$ along the $\mathbf{c}^*$ (out-of-plane) direction. The magnetization data represented by the dash lines (0 T to 30 T) are very noisy because of the huge switching electromagnetic noise inevitably generated for injection mega-ampere driving currents at the beginning of the destructive ultra-high field generation[53]. The magnetization process and d$M$/d$H$ are precisely measured from 30 to 95 T, which shows two peaks labeled by $H_c^l$ and $H_c^h$. To be specific, we have con-ducted three independent measurements (i), (ii), and (iii) in Fig. 1,

where $H_c^l$ is found to be about 35 T in three measurements (we also note that the ~ 35 T signal was not observed in the previous magneti-zation measurement[18,27], it maybe caused by the increasing ABAB stacking fault in $\alpha$-RuCl$_3$), and in agreement with the magneto-torque probe result (32.5 T)[44] (marked with the vertical dashed line in Fig. 1). On the other hand, the measured $H_c^h$ fields are somewhat different in cases (i), (ii), and (iii), with values of 76 T, 83 T, and 87 T, respectively. This difference can be attributed to the small angle ambiguity ($\pm 2.5°$) in the three measurements and also to the high sensitivity of the transition field for the field angle near the $\mathbf{c}^*$ axis of the compound[46]. Moreover, we average the d$M$/d$H$ curves from experiments (i-iii), show the results in Fig. 1d, and find the averaging process has significantly reduced the electrical noise. This allows us to identify more clearly the two peaks at $H_c^l$ and $H_c^h$, respectively.

Figure 2 shows the measured d$M$/d$H$ results for various tilting angles ranging from $\theta \simeq 0°$ (i.e., out-of-plane fields) to 90° (in-plane). For $\theta \simeq 0°$ and 9°, the data are obtained by the destructive method, while the d$M$/d$H$ curves with $\theta \simeq 20°$, 30° and 90° are obtained by the non-destructive magnet and up to about 30 T.

The three $\theta \simeq 0°$ cases are also plotted in Fig. 2. Here only the high-quality data above 30 T are shown, which exhibit double peaks at $H_c^l$ and $H_c^h$. With the single-turn coil technique reaching the ultra-high magnetic field of 100 T class, here we are able to reach the higher transition field near $H_c^h \simeq 83$ T that has not been reached before. It is noteworthy that although the down-sweep data in the field-decreasing measurements are unavailable to be integrated due to the field inhomogeneity[54,55], nevertheless the signals at $H_c^l$ and $H_c^h$ in the up-sweep and down-sweep processes are consistent (c.f., Supplementary Fig. 4). This indicates unambiguously that these two anomalies are not artifacts due to noise but genuine features of phase transitions in $\alpha$-RuCl$_3$, and the possibility that the sample becomes degraded by applying the ultrahigh field can be excluded.

At $\theta \simeq 9°$ and 20°, the signals in d$M$/d$H$ curve becomes rather weak (see also Fig. 3) although we measure the data at 9° by employing the more sensitive pick-up coil with 1.4 mm diameter. The high-field downturn feature of the curve at 9° is thought to reflect the saturation of the magnetization as field increases. To see the transition for clarity, we show the averaged d$M$/d$H$ curves measured by the non-destructive magnet in the two middle insets of Fig. 2, where round-peak signals are observed near 25 and 20 T for $\theta \simeq 9°$ and 20°. These round peaks in the middle-inset of Fig. 2 are thought to be the phase transitions. The two dome structures of averaged d$M$/d$H$ curves at 9° leads to an uncer-tainty in $\theta$ of $\pm 2.5°$. We note that the two transition fields ($H_c^l$ and $H_c^h$) for $\theta \simeq 0°$ seem to merge into one, and as this two curves are averaged results with $\theta \simeq 9 \pm 2.5°$ and $20 \pm 2.5°$, the peaks are very broad. Therefore, we define a large error bar, i.e., $\pm 5$ T for $\theta \simeq 9°$, and $\pm 2$ T for $\theta \simeq 20°$.

The results at larger angles $\theta \simeq 30°$ and 90° are also shown in Fig. 2[44,52]. The d$M$/d$H$ curve at 90° shows two peaks and one shoulder structures. The peaks at 6.2 T and 7.2 T correspond respectively to the transition boundaries of the magnetic zigzag order (zigzag1) and another zigzag order (zigzag2), in agreement to previous studies[42,45]. The shoulder structure seen at around 8.5 T is likely due to another antiferromagnetic (AFM) order[56]. Because this feature is insensitive to the field angle as we show in the latter part, such AFM order is deemed to be caused by the ABAB stacking components and the transition field is denoted as $H_c^{AB}$. For large angles, the critical field $H_c$, e.g., $H_c \simeq 7.2$ T for $\theta \simeq 90°$ (i.e., in-plane), labels the upper boundary between the zigzag and paramagnetic phases. Such a transition has been widely recognized for the in-plane case as observed by neutron scattering experiments[18,20,33], and for tilted angles based on the magneto-torque measurements[44]. Besides, the additional peak at 6.2 T is generally believed to reflect the transition between two different zigzag antiferromagnetic phases, with period-3 and period-6 spin structures along the $\mathbf{c}^*$

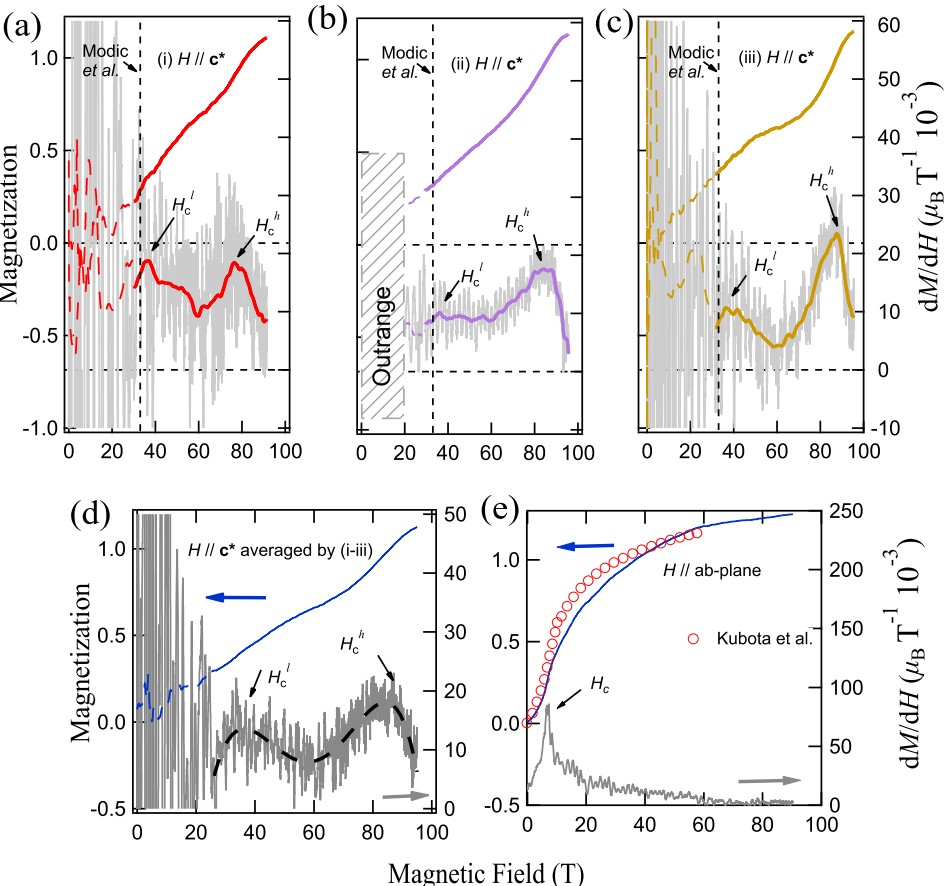

**Fig. 1 | The magnetization process of α-RuCl₃ up to 100 T. a–c** The d$M$/d$H$ data (lower) measured up to 100 T under out-of-plane fields ($H \parallel \mathbf{c}^*$) and the integrated magnetization curves (upper). The grey noisy curves are the raw d$M$/d$H$ data, with the smoothed lines also presented. The data from 0 to 30 T is shown as dash line because of the strong starting switch noise[53,54]. (i), (ii) and (iii) represent three independent experiments showing similar results despite an uncertainty in field angles of ± 2.5°, and experiment (ii) is performed with the high-frequency-cut filters. The shadow range (≤ 20 T) in (ii) is not precisely measured because of the outranged noise. The transition field along $\mathbf{c}^*$ reported by Modic et al.[44] is also marked by the vertical dashed line. **d** The averaged d$M$/d$H$ and $M-H$ data from experiment (i), (ii), and (iii), where the two phase-transition signals can be more clearly seen. The black dashed line is a guide for the eye. **e** The high-field magnetization measurements under in-plane fields up to 90 T, where only a single transition at about 7 T is observed, in excellent agreement with previous measurements by Kubota et al. (ref. 27).

direction in the ABC stacking, respectively, (see, e.g., ref. 45). Here we dub this transition field as $H'_c$.

$H_c$ and $H'_c$ are found to monotonously increase with decreasing the field angle. In contrast, $H_c^{AB}$ is independent of the field angle, suggesting that $H_c^{AB}$ at 8.5 T comes from a magnetically isotropic origin which is different from the transitions at $H_c$ and $H'_c$. According to the previous study[18,56], the field location of 8.5 T indicates that the transition occurs in the stacking fault ABAB layers in the sample. Therefore, the phase transition due to the suppression of the antiferromagnetic order in the ABAB stacking component is found to be isotropic, suggesting a 3-dimensional order which is different from the 2-dimensional zigzag orders.

Here, we should note that the presence of ABAB stacking fault is almost inevitable for α-RuCl₃ in the out-of-plane high magnetic field experiment. This is because the stress caused by the strong magnetic anisotropy under the magnetic field along the $\mathbf{c}^*$ axis would more or less deform the sample[57]. We can even damage α-RuCl₃ by deforming the sample and produce lots of ABAB stacking faults, which now exhibits ordering temperature at about 14 K (c.f., Supplementary Fig. 5). Then we perform high-field experiments up to 100 T along the $\mathbf{c}^*$ axis on this sample, and find only $H_c^{AB}$ peak at around 14(± 4) T. Based on the experimental results, we conclude that the $H_c^l$ and $H_c^h$ signals should belong to the ABC stacking component. Furthermore, we also note that the pulse time of the destructive magnet is only a few

microseconds[55], much shorter as compared to the non-destructive magnet. This allows the samples to withstand less stress impulse during the measurement, rendering some advantages in measuring fragile and strong anisotropic samples such as α-RuCl₃.

In Fig. 2, by comparing the d$M$/d$H$ results at different $\theta$ angles from 90° to 0°, we find strong magnetic anisotropy consistent with previous measurements[18,27]. We measured the magnetization process for $\theta \simeq 90°$ (within the $ab$-plane) up to 90 T using the single-turn coil techniques. The results are shown in Fig. 1e, which demonstrate that only the 7 T transition is present for $\theta \simeq 90°$ and our measurements reproduce excellently the results in ref. 27 [c.f., Fig. 1 e]. It is found that $H_c$ monotonically increases with decreasing angle from 90° to 0°, which is consistent with the results of Modic et al.[44]

As we described in Fig. 1, the d$M$/d$H$ at 0° is significantly different from that at large angles ($\theta \gtrsim 9°$) and exhibits two phase transitions. The two phase transitions indicate that an intermediate phase emerges between $H_c^l$ and $H_c^h$. Because Modic et al.[44] have claimed that the zigzag order is suppressed for $H > H_c$ or $H'_c$, the intermediate phase between $H_c^l$ and $H_c^h$ should be disordered and counts as the experimental evidence of the recently proposed QSL phase in α-RuCl₃ with fields applied along out-of-plane $\mathbf{c}^*$ axis[47,51]. We also note that there is another scenario that $H_c^h$ corresponds to the transition field that suppresses the AFM order, and $H_c^l$ just separates two different AFM phases. However, based on the experimental

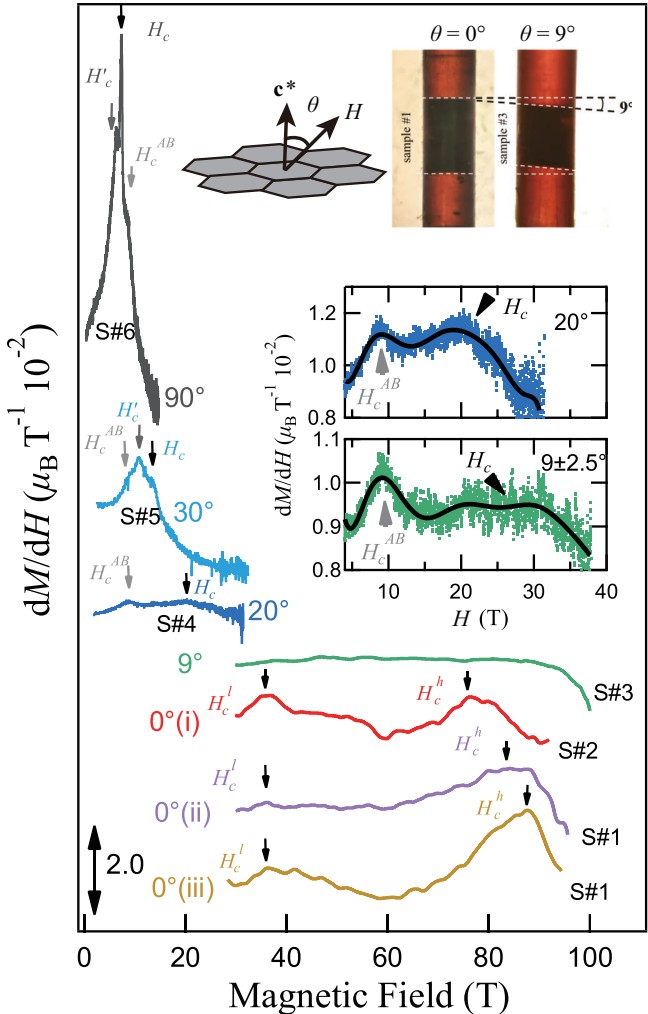

**Fig. 2 | The d$M$/d$H$ curves at various $\theta$ angles.** We include the measurements with $\theta \simeq 0°, 9°, 20°, 30°, 90°$, where the $0°$ measurements are performed for multiple times (NoS. i, ii, and iii) using the destructive method with possible tilting angle within $\pm 2.5°$. Sample #1-6 represent the sample number in different field directions (S#1-6). The black arrows pointing to the peaks of d$M$/d$H$ denote the transition fields in the measurements, while the grey ones with $H_c'$ and $H_c^{AB}$ indicate the irrelevant feature due to the three dimensional spin structure[45] and the magnetic phase transition in sample with ABAB stacking fault, respectively. The upper inset illustrates the angle $\theta$ between the applied magnetic field and $\mathbf{c}^*$ axis, as well as the photos of holding setup of the samples for $\theta \simeq 0°$ and $9°$. The two middle insets show the averaged d$M$/d$H$ curves obtained by the non-destructive magnet because the transition signals are very weak. The black solid curves are guides for the eye.

results here, the reported data of Modic et al.[44], and calculated results as shown in the following section, we find strong evidence that the transition at $H_c^l$ is an intrinsic characteristic of the ABC stacking component, and consider it is more reasonable that $H_c^l$ suppresses the AFM order of the ABC stacking sample.

## Comparison between experimental and calculated results

The recently proposed realistic microscopic spin model with large Kitaev coupling might support our experimental results. We consider the $K$-$J$-$\Gamma$-$\Gamma'$ model $\mathcal{H}_0 = \sum_{\langle i,j \rangle_\gamma} [K S_i^\gamma S_j^\gamma + J \mathbf{S}_i \cdot \mathbf{S}_j + \Gamma(S_i^\alpha S_j^\beta + S_i^\beta S_j^\alpha) + \Gamma'(S_i^\gamma S_j^\alpha + S_i^\gamma S_j^\beta + S_i^\alpha S_j^\gamma + S_i^\beta S_j^\gamma)]$ ($\alpha, \beta, \gamma \in \{x, y, z\}$) with parameters $K = -25$ meV, $J = -0.1|K|$, $\Gamma = 0.3|K|$, and $\Gamma' = -0.02|K|$[51].

In Fig. 3, we compare the experimental and calculated d$M$/d$H$ results as well as the integrated $M$-$H$ results. For the experimental data, only the critical fields associated with the pristine ABC stacking component, i.e., $H_c$, $H_c^l$, and $H_c^h$, are marked.

From Fig. 3a, b, we find semi-quantitative agreement between the experimental and calculated d$M$/d$H$ results. Similarly, the experimental and calculated $M$-$H$ results also show consistency to each other as shown in Fig. 3c, d. In Fig. 3b, for small angles $\theta = 0°, 0.8°$, and $2.0°$ located within the angle range $\theta \simeq 0° \pm 2.5°$, the calculated curves exhibit two transition fields as indicated by the solid black triangles and circles, and we find the upper transition fields $H_c^h$ are rather sensitive to the small change of $\theta$ near $0°$. Therefore it explains the visible difference in $H_c^h$ among the three $\theta \simeq 0°$ measurements. On the contrary, the lower transition field $H_c^l$ is found rather stable in Fig. 3b, also in agreement with experiments. As the angle $\theta$ further increases, e.g., $\theta = 10°$, there exists a single transition field, in agreement with the experimental result of $9°$ in Fig. 3a. The calculated transition fields $H_c$, from our DMRG simulations based on the 2D spin model, of $\theta \simeq 20°, 30°$, and $90°$ cases in Fig. 3b show quantitative agreement to measurements in Fig. 3a. We note that there are still certain differences between the DMRG and experimental results, such as the height of peaks, which are understandable. The difference might be ascribed to the finite-size effects in the model calculations (c.f., Supplementary Fig. 7) or other possible terms/factors not considered in the present model study, e.g., the next- and third-nearest neighbor Heisenberg couplings, the inter-layer interactions, and the inhomogeneous external field in the high-field measurements. In particular, as the DMRG calculations are performed on an effective two-dimensional spin model, the inter-layer stacking effects in $\alpha$-RuCl$_3$ compounds are not considered.

## Discussion

From both experimental and calculated magnetization data, we see intrinsic angle dependence of the quantum spin states in $\alpha$-RuCl$_3$ under magnetic fields. Therefore, by collecting the transition fields $H_c^l$ and $H_c^h$ marked in Fig. 3, we summarize the results in a field-angle phase diagram shown in Fig. 4. In previous theoretical studies, an intermediate QSL phase was predicted between the upper boundary of zigzag phase $H_c^l$ and the lower boundary of paramagnetic phase $H_c^h$[47,51]. Nevertheless, the fate of the intermediate QSL phase under tilted angles has not been studied before. Here we show clearly that the QSL states indeed constitute an extended phase in the field-angle phase diagram in Fig. 4, as further supported by additional DMRG calculations of the spin structure factors here (c.f. the Supplementary Section B). Moreover, when $\theta$ becomes greater than about $9°$, there exists only one transition field $H_c$ in Fig. 4, which decreases monotonically as $\theta$ further increases. The previously proposed magnetic transition points determined by the magneto-torque measurements[44] are also plotted in Fig. 4 and found to agree with our $H_c$ for $\theta$ from $9°$ to $90°$. In addition, the two transitions ($H_c^l$ and $H_c^h$) experimentally obtained at $\theta \simeq 0°$ are semiquantitatively reproduced by the theoretical simulation, which indicates the existence of an intermediate QSL phase. The transition field of the magneto-torque measurements[44] at $\theta = 0°$ is also found to be in agreement with our results. For $0° < \theta \lesssim 9°$, there is a discrepancy between the theoretical simulation and the results of the torque measurement. Although the reason of the difference is not completely clear at present, the quantum fluctuations in the vicinity of the potential tricritical point where the $H_c^l$ and $H_c^h$ merge disturbs the precise evaluation of the transition field experimentally as well as numerically. Nevertheless, the theoretical proposition of the extension of the QSL phase to the finite small $\theta$ is likely to be supported by different experimental $H_c^h$ at $\theta \simeq 0°$ with $\pm 2.5°$ uncertainty.

In summary, we find experimentally an interesting two-transition scenario in the prime Kitaev material $\alpha$-RuCl$_3$ under high out-of-plane fields up to 100-T class and reveal the existence of a field-induced intermediate phase in the field-angle phase diagram. Such a magnetic disordered phase is separated from the trivial polarized state by a quantum phase transition, suggesting the existence of the long-sought QSL phase as predicted in previous model

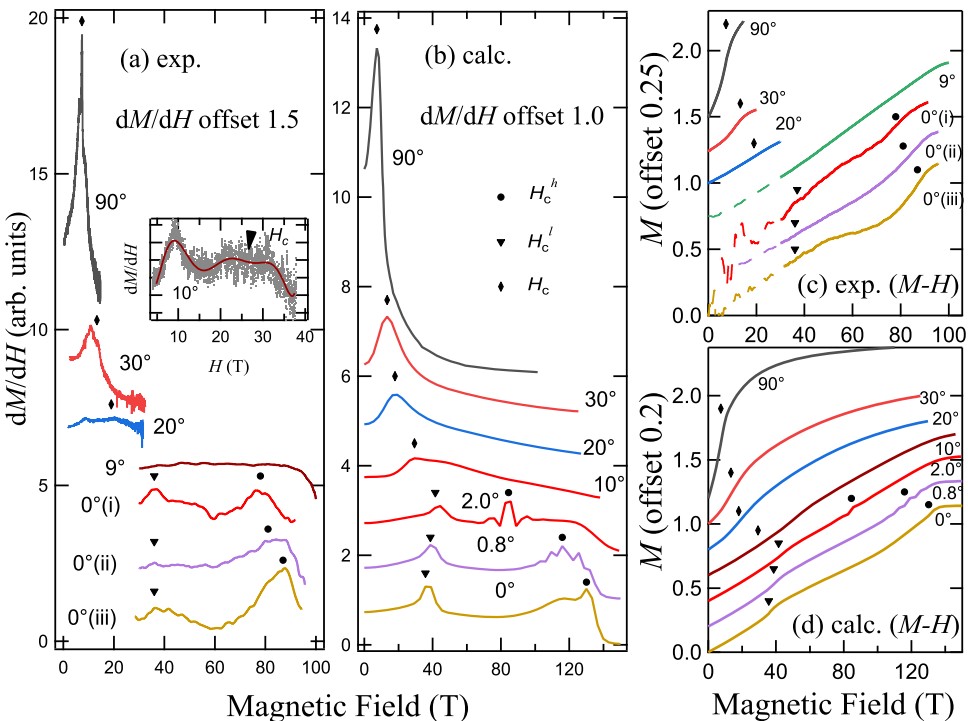

**Fig. 3 | Comparison between the experimental and calculated results. a** The experimental d$M$/d$H$ data and **b** the calculated results for various $\theta$ angles. **c** The integrated $M$-$H$ curves as well as **d** the calculated results. The markers of diamond, triangle, and circle denotes $H_c$, $H_c^l$, and $H_c^h$, respectively. The experimental transition field at 10° are labeled in the inset of **a**. Some calculated results in other $\theta$ angles are shown in Supplementary Section B. The $M$-$H$ data of destructive measurements below 30 T are represented by dash lines.

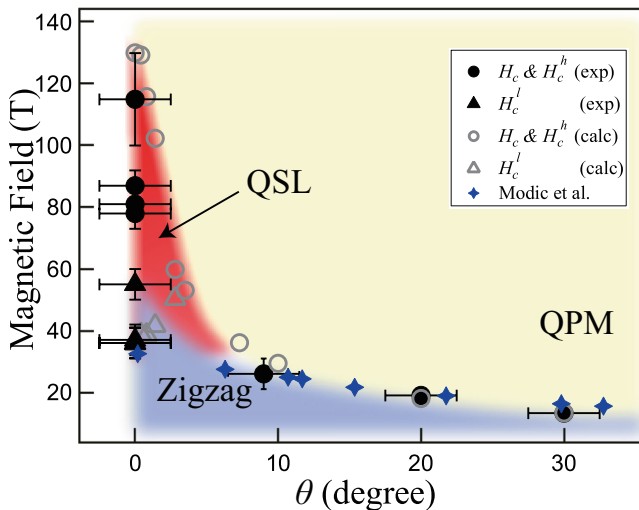

**Fig. 4 | The field-angle phase diagram.** The field-angle phase diagram that summarizes the values of transition fields determined from both the experimental (black solid markers) and the calculated (grey open ones) $H_c$, $H_c^l$, and $H_c^h$. We also plot the low-field results (blue stars) taken from ref. 44 as a supplement. The zigzag antiferromagnetic, quantum paramagnetic (QPM), and the quantum spin liquid (QSL) phases are indicated.

studies[47,51]. Regarding the nature of the intermediate QSL phase, previous theoretical work[47] concludes the intermediate QSL phase can be adiabatically connected to the Kitaev spin liquid (KSL) phase. On the other hand, ref. 51 draws a different conclusion of gapless QSL in the intermediate regime based on results with multiple many-body approaches. Here we further uncover that the intermediate phase also extends to a finite-angle regime, whose precise nature calls for further theoretical studies. While the phase diagram in Fig. 4 excludes the presence of an in-plane QSL phase like certain other recent studies[43,44], our work nevertheless opens the avenue for the exploration of the out-of-plane QSL phase in the Kitaev materials. Moreover, further experimental characteristics of the intermediate QSL phase can be started from here. For example, nuclear magnetic resonance and electron spin resonance spectroscopy under high fields[58,59] are promising approaches for probing low-energy excitations in the intermediate QSL phase discovered here.

## Methods

### Experimental details

A single crystal of $\alpha$-RuCl$_3$ was used for the present experiment[27]. The vertical-type single-turn coil field generator was employed to provide a pulse magnetic field up to 102 T. Things inside of the coil including the sample are generally not damaged by the generation of a magnetic field, although the field generation is destructive[55]. The magnetization processes under the out-of-plane fields (Fig. 1) and those with small tilting angles (9° lines in Fig. 2) were measured using a double-layer pick-up coil that consists of two small coils compensating for each other[53,55]. The sample is cut to 0.9 × 0.9 mm$^2$ square. Several sample with ∼0.2 mm thickness are stacking together to obtain enough thickness to measure the magnetization process in the single-turn coil experiments. The angle between the magnetic field and the **c**$^*$ axis is denoted as $\theta$ (c.f. upper inset of Fig. 2). In order to have good control on the angle $\theta$, two glass rods with a section inclination angle $\theta$ are employed to clamp the sample in a Kapton tube. The single-turn coil, pick-up coil, and the Kapton tube with the sample are placed in parallel visually. As the $\alpha$-RuCl$_3$ sample is very soft and has strong anisotropy, it needs to be carefully fixed. Silicone grease instead of cryogenic glue is used to hold the sample, in order to reduce the dislocation of stacking caused by pressure (For more information of the set-up around the sample, see in Supplementary Fig. 3). Nevertheless, such an experimental setting inevitably affects the precise control of $\theta$ with errors estimated to be ±2.5°.

Two types of double-layer pick-up coils are employed in the measurements; one is the standard type with 1 mm diameter[53], and the other is a recently developed one with a larger diameter of 1.4 mm that helps to enhance the signal by nearly three times. The magnetization signal is obtained by subtraction of the background signal from the sample signal, which are obtained by two successive destructive-field measurements[53–55] without and with the sample (see Supplementary Fig. 3), respectively. Magnetization measurements at certain large angles like $\theta \simeq 9°, 20°, 30°$, and $90°$ are performed by a similar induction method employing non-destructive pulse magnets[60]. In the non-destructive pulse field experiment, the diameter of the sample is about 2 mm. All of our experiments are performed at a low temperature of 4.2 K.

### Density matrix renormalization group calculation

We simulate the system on the cylindrical geometry up to width 6 (c.f. Supplementary Sec. B), and retain $D = 512$ bond states that lead to accurate results (truncation errors less than $\epsilon \simeq 1 \times 10^{-6}$). The direction of the magnetic field $H$ is represented by $[lmn]$ in the spin space ($S^x, S^y, S^z$), and the Zeeman term reads $\mathcal{H}_H = g\mu_B \mu_0 H_{[lmn]} \frac{lS^x + mS^y + nS^z}{\sqrt{l^2 + m^2 + n^2}}$ with $H_{[l=1, m=1, n]}$ tilting an angle $\theta = \arccos\left(\frac{2+n}{\sqrt{6+3n^2}}\right) \cdot \frac{180°}{\pi}$ to the $\mathbf{c}^*$ axis within the $a\mathbf{c}^*$-plane, and the Landé $g$-factor is fixed as $g \simeq 2.3$. The magnetization curves shown in Fig. 3b are obtained by computing $M = g\mu_B \frac{l\langle S^x \rangle + m\langle S^y \rangle + n\langle S^z \rangle}{\sqrt{l^2 + m^2 + n^2}}$.

## Data availability

The data that support the findings of this study are available from the corresponding author upon reasonable request.

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

## Acknowledgements
X.-G.Z. thank Yuan Yao for fruitful discussions, and acknowledge Yuto Ishi, Hironobu Sawabe, and Akihiko Ikeda for experimental supports. W.L. and H.L. are indebted to Shun-Yao Yu, Shou-Shu Gong, Zheng-Xin Liu, and Jinsheng Wen for helpful discussions. X.-G.Z was supported by a JSPS fellowship. X.-G.Z. and Y.M.H. was funded by JSPS KAKENHI No. 22F22332. Y.H.M. was funded by JSPS KAKENHI, Grant-in-Aid for Transformative Research Areas (A) Nos.23H04859 and 23H04860, Grant-in-Aid for Scientific Research (B) No. 23H01117, and Grant-in-Aid for Challenging Research (Pioneering) No.20K20521. H.L. and W.L. were supported by the National Natural Science Foundation of China (Grant Nos. 12222412, 11834014, 11974036, and 12047503), CAS Project for Young Scientists in Basic Research (Grant No. YSBR-057), and China National Postdoctoral Program for Innovative Talents (Grant No. BX20220291).

## Author contributions
Y.H.M and W.L supervised the project. X.-G.Z and Y.H.M performed the destructive magnetic field experiment. X.-G.Z, A.M and K.K performed the non-destructive field experiment. N.K and H.T provided the sample α-RuCl$_3$. X.-G.Z and Y.H.M analyzed the experimental data. H.L, W.L and G.S performed the model calculations and analyzed the numerical results. X.-G.Z, H.L, Y.H.M and W.L contributed to the preparation of the draft.

## Competing interests
The authors declare no competing interests.
