## [Peer Review File · Nature Communications]

REVIEWER COMMENTS

Reviewer #1 (Remarks to the Author):

Authors investigated possible phase transitions in α - RuCl_3 , a promising Kitaev candidate material under the magnetic field. In particular, they studied the c -axis (out-of-plane) field where the field-induced intermediate phase was predicted using the spin exchange parameters relevant for α - RuCl_3 . The previous theoretical work also suggested that this intermediate disordered phase (putative quantum spin liquid) quickly disappears when the field is tilted out of the plane by ~ 10 degrees.

Authors reported the two transitions where the intermediate phase is indeed bound by two critical fields, h_c^l and h_c^h , while no such phase is found when the field is parallel to the in-plane direction, consistent with the theoretical results.

I found this work valuable for the quantum magnetism community. There are a few experimental works (published in nature journals) reporting a possible quantum spin liquid when the field is applied along the a -axis (which is parallel to the zig-zag chain made of honeycomb bond), while none of the theoretical work is able to find such phase. On the other hand, the out-of-plane field induced disordered phase was reported by several different numerical techniques as authors listed in their references.

I have one comment which is not directly related, but would be important, as the author may mislead readers. In the discussion, authors stated "... out-of-plane QSL phase... not limited to α - RuCl_3 , but may also apply to ... like $\text{Na}_2\text{Co}_2\text{TeO}_6$ (NCTO) and $\text{BaCo}_2(\text{AsO}_4)_2$ (BCAO)." This is not the case, since the out-of-plane QSL is due to the significant AFM Γ term, which is important even though Kitaev is the largest interaction in α - RuCl_3 ; the AFM Γ gives a strong in-plane vs. out-of-plane anisotropy, and also generates this intermediate QSL. For Co^{2+} , it is known that Γ and Γ' are tiny, and the anisotropy is due to g -factor anisotropy (i.e, trigonal distortion + spin orbit coupling). Thus, we don't expect the out-of-plane QSL in honeycomb cobaltates. There are many competing interactions such as J_1, J_3 , in addition to the Kitaev interaction in $3d^7$ materials. I would suggest that the author either remove the sentence or revise it with full information and relevant references if they decide to keep it.

Overall, the work reported in this manuscript will advance our search for QSL, even though the nature of the intermediate phase still remains to be resolved. I recommend the publication in Nature Communications.

Reviewer #2 (Remarks to the Author):

Based on recent theoretical studies, measurements of RuCl₃ with magnetic field applied near or along the c-axis is well-motivated. Several works using various advanced numerical methods claim that a spin liquid state is likely for these field orientations and this regime has not been explored in much detail prior to this work. Thus, this data is interesting and of value to the community. However, these experiments are certainly challenging, especially given the delicate nature of the RuCl₃ samples.

The main issue with this work is the lack of reproducibility of the data in comparison to the literature, which will not convince readers of the high field transition. The data does not reproduce known data in the field range where H_c has already been measured (i.e. at 9 degrees in Figure 4, H_c^l is not shown and H_c^h is much larger than the H_c observed by Modic et.al.). I am confused why the authors need to use labels of H_c ("the upper boundary of the zigzag phase") and H_c^l because presumably they should be describing the same thing---the suppression of AFM order. If data existed below ~30 tesla in the 9 degree curve and H_c were clearly shown and in agreement with the literature, this would help convince readers that there are in fact two transitions in RuCl₃ as a function of magnetic field, one that corresponds to the suppression of AFM order and another that might correspond to the suppression of a QSL state at higher fields. This would represent an important finding, however, even the data near 0 degrees is not very convincing of the fact that there are two transitions.

I think it's a bit problematic that the downsweep data cannot be used and the data below 30 T on the upsweep also cannot be used. I understand that each really high field pulse is a lot of effort, but the angle dependence of the lower H_c (the one describing the suppression of AFM order) needs to be clearly demonstrated for some angles near zero degrees while maintaining that H_c is still above the noisy region of 30 tesla. Unfortunately, almost all field orientations (apart from those very near to the c^* axis) have an H_c below 30 tesla and none of these field orientations were measured up to 100 tesla to confirm that there are two transitions in the same pulse. The noise in the raw data (gray curves in Figure 1) is very large and the evidence for a transition near 35 tesla is no larger in magnitude than several peaks observed at lower fields. Furthermore, one could argue that the 35 tesla transition is not even observed in experiment ii.

One should always assume that there is some precession about the high symmetry directions of the crystal when rotating with respect to magnetic field. Thus, I could easily imagine that even the H_c as determined by Modic et.al. is not reaching the maximum value that one expects at c^* , and it could very well be that the feature that the authors observe ~ 80 tesla is in fact the suppression of AFM order. This is reflected in the fact that H_c increases significantly as one approaches the c^* direction (Figure 1b in reference 44) and even a small misalignment by Modic et.al. could reduce this value from 80 tesla to the ~ 35 tesla value shown in Figure 1b.

Another point of concern is the H_c^{AB} transition, which the authors note is irrelevant. I think observing this transition suggests that a significant fraction of the sample has been damaged (gray arrow in Figure S1), possibly during handling or preparing the samples, or by the large torques experienced in a magnetic field. I agree with the authors that when a large portion of the sample experiences ABAB stacking, the system becomes more isotropic and thus, one might expect the H_c^{AB} transition not to change much with angle. However, there are several concerning aspects about the observed anisotropy (or lack thereof) in other transitions. For example, can the authors explain why H_c^A in the near zero degree data is fairly isotropic in experiments i-iii? I presume H_c^A corresponds to the AFM transition observed by others, which we know is highly anisotropic. This weakens evidence that the H_c^A transition is the AFM transition---either the data is too noisy and switching noise is introducing additional features or the authors do not understand the origin of H_c^A . It does not help that the “irrelevant” H_c^{AB} transition (down-sweep data in Figure S4) is as large or larger than the H_c^A and H_c^h transitions that the authors claim is where readers should focus their attention.

In summary, I cannot recommend this paper for publication as I think the experiments need to be better executed and the data is not convincing. I commend the authors on their efforts to obtain such a challenging dataset, but the lower inset in Figure 2 summarizes the fact that the authors cannot distinguish real transitions from the electrical noise in their measurement. I actually agree with the authors that it looks like there may be evidence for a transition near 80 tesla, but one should first confirm other known transitions at lower fields and thereby confirm that the 80 tesla transition is not the suppression of AFM order. I appreciate that other methods to study RuCl_3 in such high magnetic fields is limited, but without complementary information and with their current data as-is, it is not sufficient to claim that the intermediate field regime between H_c^A and H_c^h is a quantum spin liquid.

I hope that these measurements can be improved to reduce noise and substantiate the authors' claim in the future.

Minor comments (some of these are repetitive to that written above, but sometimes it's useful to see it said in different words):

- Line 32, renowned honeycomb model “with bond-dependent exchange”
- Line 45, chlorine?
- I don’t understand the last sentence of the caption for figure 2
- What is the reason for the downturn at the highest fields on the 9 deg data?
- Page 3, line 17-19 Why does H_c in the 9 deg data occur at higher field than H_c^{\perp} for 0 deg?
- Line 55, emergence
- Page 4, line 1-3, this summarizes the main problem with the paper because it suggests that data taken in a different magnet or at a different angle is qualitatively different, and one would expect a smooth evolution of these transitions. Furthermore, there is no overlap of consistency (or field range) between this data set and that found in the literature.
- Page 4, Line5-10, just because it’s disordered above H_c , doesn’t mean it’s a spin liquid. This is insufficient to make this claim.
- Line 13; recent
- In Figure 3, the H_c^{\perp} field seems angle independent in the data in the near zero curves while H_c^h changes a lot between experiments i-iii. How can you rule out that H_c^{\perp} is not related to ABAB stacking because even the calculation suggests that it should be very sensitive to angle. Also, in the calculation it follows the wrong angle dependence compared to the literature. Do you understand this?
- Figure S4, where is the other gray curve? Why is only the derivative shown on the down sweep? The down sweep data looks less noisy so why isn’t it used in the main text?
- The main issue with Figure S4 is that the H_c^{\perp} transition is less pronounced than the H_c^{AB} transition. The H_c^{\perp} transition is too near the noise in the up sweep data. For angles near the c^* axis, it’s possible that the H_c^h transition is the H_c associated with AFM.
- My apologies if I missed this in the manuscript, but what is the approximate size of the samples being measured?

Reviewer #3 (Remarks to the Author):

In this manuscript Zhou et al. measured the magnetization of RuCl_3 under rotatable magnetic fields up to 95 T at 4.2 K. This wide field range is enabled by a specialized single-turn coil technique

developed by the group over the past years. Compared with previous reports (e.g., Nat. Phys. 17, 240 (2021)), the major novelty of this manuscript is the observation of two magnetic transitions (H_{cl} and H_{ch}) under an out-of-plane field. The region of interest is in the middle of the two transitions ($H_{cl} < H < H_{ch}$), where DMRG simulations show that the zigzag magnetic order is suppressed by the field yet it is not strong enough to fully polarize the spins. This was taken as a hint of a possible quantum spin liquid phase. By rotating the field away from the out-of-plane direction, the two transitions seemingly merge into one, with the transition field shifting downwards in strength. Reassuringly, the magnetization of RuCl_3 measured in this manuscript under an in-plane field is similar to previous reports.

Overall, I find the results quite interesting and the experiments carefully performed. The manuscript will surely add to the interest in a material system that has already received great attention. Therefore, I recommend the publication of the manuscript in Nature Communications, provided that the following comments are properly addressed.

1. The data measured with field angles between 0° and 30° are not very convincing. Especially, the 9° and 20° magnetization curves have no obvious peaks in my opinion. I noticed that the 9° data was measured on a different sample with a different pickup coil. Can the authors provide more and higher quality data in this field range? I find it difficult to conclude that the two transition fields (H_{cl} , H_{ch}) at 0° merge into one transition at 9° from the data currently presented in Fig. 2, yet this is crucial for the interpretation of the data. In addition, when comparing to the DMRG calculations in Fig. 3, the experimental peaks at 9° and 20° are also much weaker than the theoretical counterparts.

2. Since most research interest in RuCl_3 is related to the magnetic behavior under an in-plane (rather than out-of-plane) field, it would certainly help the manuscript to show a high-in-plane-field magnetization curve using this special single-coil technique. This measurement is also important as a cross-reference to the out-of-plane curves.

3. Can the authors provide a physical picture of why the two magnetic transitions appear as peaks in dM/dH curves? It is not very intuitive how to reconcile the magnetization slopes with the change in the zigzag/other magnetic orders.

4. Other minor points:

- It is not easy to distinguish the M-H and dM/dH -H curves in Fig.1. Please use arrows to indicate the two sets of curves (similar to the fashion in Fig. S1).
- On page 2 of the SI, “note” should be “not”.
- On line 14, page 4 of the main text, “resent” should be “recently”. In addition, it is unclear what “high K-term” means.
- “(0, 30) T” in the last sentence of the caption to Fig. 3 is unclear. Please rephrase.

RESPONSE TO THE FIRST REVIEWER'S REPORT

We thank the Reviewer #1 for the high praise of our work and the recommendation of its publication. The constructive suggestion also helped us to further improve our manuscript.

Reviewer #1-summary : *Authors investigated possible phase transitions in α -RuCl₃, a promising Kitaev candidate material under the magnetic field. In particular, they studied the c-axis (out-of-plane) field where the field-induced intermediate phase was predicted using the spin exchange parameters relevant for α -RuCl₃. The previous theoretical work also suggested that this intermediate disordered phase (putative quantum spin liquid) quickly disappears when the field is tilted out of the plane by ~ 10 degrees.*

Authors reported the two transitions where the intermediate phase is indeed bound by two critical fields, h_c^l and h_c^h , while no such phase is found when the field is parallel to the in-plane direction, consistent with the theoretical results.

I found this work valuable for the quantum magnetism community. There are a few experimental works (published in nature journals) reporting a possible quantum spin liquid when the field is applied along the a-axis (which is parallel to the zig-zag chain made of honeycomb bond), while none of the theoretical work is able to find such phase. On the other hand, the out-of-plane field induced disordered phase was reported by several different numerical techniques as authors listed in their references.

Reply: We thank the Reviewer for the precise summary and high assessment of our work. Indeed, the field-induced quantum spin liquid (QSL) states have attracted great attention both in theoretical and experimental studies. In a previous model study, some of the coauthors (HL and WL) found a realistic model for α -RuCl₃ and predicted the presence of an intermediate QSL phase under high magnetic fields. Here through the very challenging high-field experiments, we not only found evidence for the QSL but also explored the comprehensive field-angle quantum phase diagram for the first time. Our results lay down the foundation for understanding the intriguing field-induced quantum spin states and transitions in the renowned Kitaev material α -RuCl₃.

Reviewer #1-1 : *I have one comment which is not directly related, but would be important, as the author may mislead readers. In the discussion, authors stated "... out-of-plane QSL phase... not limited to α -RuCl₃, but may also apply to ... like Na₂Co₂TeO₆ (NCTO) and BaCo₂(AsO₄)₂ (BCAO)." This is not the case, since the out-of-plane QSL is due to the significant AFM Gamma term, which is important even though Kitaev is the largest interaction in alpha-RuCl₃; the AFM Gamma gives a strong in-plane vs. out-of-plane anisotropy, and also generates this intermediate QSL. For Co²⁺, it is known that Gamma and Gamma' are tiny, and the anisotropy is due to g-factor anisotropy (i.e, trigonal distortion + spin orbit coupling). Thus, we don't expect the out-of-plane QSL in honeycomb cobaltates. There are many competing interactions such as J1, J3, in addition to the Kitaev interaction in 3d7 materials. I would suggest that the author either remove the sentence or revise it with full information and relevant references if they decide to keep it.*

Reply: We thank the reviewer for this insightful comment and suggestion. Indeed, the Γ term plays an important role for the emergence of intermediate QSL in α -RuCl₃ under out-of-plane fields. As the Reviewer suggested, given the Γ and Γ' are tiny in Co-based Kitaev candidates, the situation there may be very different and the argument by analogue is not proper any more. Therefore, we have removed the related discussions in the revised manuscript.

Reviewer #1-2 : *Overall, the work reported in this manuscript will advance our search for QSL, even though the nature of the intermediate phase still remains to be resolved. I recommend the publication in Nature Communications.*

Reply: We thank the Reviewer for the recommendation of publication, with the belief that our work can further stimulate high-field studies of α -RuCl₃ and similar Kitaev materials.

RESPONSE TO THE SECOND REVIEWER'S REPORT

Reviewer #2-summary : *Based on recent theoretical studies, measurements of α -RuCl₃ with magnetic field applied near or along the c -axis is well-motivated. Several works using various advanced numerical methods claim that a spin liquid state is likely for these field orientations and this regime has not been explored in much detail prior to this work. Thus, this data is interesting and of value to the community. However, these experiments are certainly challenging, especially given the delicate nature of the α -RuCl₃ samples.*

Reply: Thanks for the concise summary. Indeed, as the reviewer mentioned, the high-field experiment is very motivated yet meanwhile very challenging. In this work, we overcome the technical difficulties for high-field magnetization measurements up to 100 T and manage to find evidences of two spin state transitions enclosing an intermediate phase, as predicted in certain previous numerical works. In the point-by-point response below, we will provide further experimental data and address the Reviewer's concerns in details.

Reviewer #2-1 : *The main issue with this work is the lack of reproducibility of the data in comparison to the literature, which will not convince readers of the high field transition. The data does not reproduce known data in the field range where H_c has already been measured (i.e. at 9 degrees in Figure 4, H_c^l is not shown and H_c^h is much larger than the H_c observed by Modic et.al.). I am confused why the authors need to use labels of H_c ("the upper boundary of the zigzag phase") and H_c^l because presumably they should be describing the same thing—the suppression of AFM order. If data existed below ~ 30 tesla in the 9 degree curve and H_c were clearly shown and in agreement with the literature, this would help convince readers that there are in fact two transitions in α -RuCl₃ as a function of magnetic field, one that corresponds to the suppression of AFM order and another that might correspond to the suppression of a QSL state at higher fields. This would represent an important finding, however, even the data near 0 degrees is not very convincing of the fact that there are two transitions.*

I think it's a bit problematic that the down sweep data cannot be used and the data below 30 T on the up sweep also cannot be used. I understand that each really high field pulse is a lot of effort, but the angle dependence of the lower H_c (the one describing the suppression of AFM order) needs to be clearly demonstrated for some angles near zero degrees while maintaining that H_c is still above the noisy region of 30 tesla. Unfortunately, almost all field orientations (apart from those very near to the c^ axis) have an H_c below 30 tesla and none of these field orientations were measured up to 100 tesla to confirm that there are two transitions in the same pulse. The noise in the raw data (gray curves in Figure 1) is very large and the evidence for a transition near 35 tesla is no larger in magnitude than several peaks observed at lower fields. Furthermore, one could argue that the 35 tesla transition is not even observed in experiment ii.*

One should always assume that there is some precession about the high symmetry directions of the crystal when rotating with respect to magnetic field. Thus, I could easily imagine that even the H_c as determined

by Modic *et.al.* is not reaching the maximum value that one expects at c^* , and it could very well be that the feature that the authors observe ~ 80 tesla is in fact the suppression of AFM order. This is reflected in the fact that H_c increases significantly as one approaches the c^* direction (Figure 1b in reference 44) and even a small misalignment by Modic *et.al.* could reduce this value from 80 tesla to the 35 tesla value shown in Figure 1b.

Reply: We thank the Reviewer for the important comments, and below we address the concern of Reviewer #2 in details.

In the high-field measurements, it is a critical challenge to observe a very weak signal. Generally, the way to address this issue is to perform the measurements for many times (such as 100 times) and then take the average to reduce the noises. However, in our single-turn coil experiment, conducting so many times of measurements is extremely challenging. Nevertheless, in practice we can perform independent experiments for a couple of times to judge whether the signals are from the electrical noise or the sample. In previous works, we have rich experiences measuring the magnetization process up to 100 T for the quantum spin systems, even when the magnetization signals are very weak (see e.g., [PRL 125 (26), 267207], [PRL 111 (13), 137204], [PRB 105 214430], [JPSJ 86 (5), 054710], [JPSJ 86 (10), 104713], and [JLTP 170 (5), 452-456]). Here, we present in Fig. R1 the dM/dH curve averaged over three independent measurements (i-iii), where the peaks corresponding to the two transitions are clear. Furthermore, the down-sweep dM/dH data also exhibit two peaks at H_c^l and H_c^h as shown in Fig. S4. Therefore, we believe the two peaks (H_c^l and H_c^h , used to label the lower- and higher-field transitions) observed in Fig. R1 are intrinsic signals reflecting the two quantum phase transitions in α -RuCl₃ driven by high magnetic fields.

FIG. R1. Averaged data over three independent measurements (i-iii), where the black dashed curve is a guide for the eye.

Here we note the concern on the signals relating to the ABAB stacking faults are left in the next section #2-2, and the two observed signals at H_c^l and H_c^h under the out-of-plane external fields faithfully reflect

FIG. R2. The averaged dM/dH curves for $\theta \simeq 9^\circ$ and 20° measured at 4.2 K by employing the non-destructive magnet. Based on this data, we revise the definition of H_c at $\theta \simeq 9^\circ$.

the transitions of ABC stacking α -RuCl₃ sample. Meanwhile, we regard the variance of H_c^h in three experiments (i-iii) in Fig. 1 is caused by the θ angle difference, i.e., $\theta \simeq 0^\circ \pm 2.5^\circ$. In order to give all of the information, we show the raw data in Fig. 1 in the manuscript. The lower field transition at H_c^l (35 T) is consistent with the previously reported results by Modic *et al.*, and the second phase transition at H_c^h is for the first time observed in experiment. As the two phase transitions can be clearly observed under out-of-plane fields, it is more likely that H_c^l suppresses the AFM order and H_c^h polarizes the system (rather than H_c^h for the suppression of AFM order). This is also consistent with previous theoretical predictions based on a realistic spin model for α -RuCl₃, and thus supports the existence of an intermediate QSL state under the out-of-plane magnetic fields.

For the concern of the Reviewer about the reproducibility of the data measured at $\theta \simeq 9^\circ$, we agree with the Reviewer that the signal of peak was too weak to fully recognise in the previously shown dM/dH curve measured at 9° . Therefore, we further perform experiments by employing the non-destructive magnet, and show the new results in Fig. R2. Based on these data, we locate more precisely $H_c \approx 25 \pm 5$ T at $9 \pm 2.5^\circ$, in agreement with Modic's results despite a recognised signal at ~ 10 T due to ABAB stacking fault (see the grey arrow with H_c^{AB} in Fig. R2). Meanwhile, we also find that the signal at 9° is overall quite weak, consistent with our previous measurement with the single-turn coil method and also model calculations. Here, we note that the small transition signal observed at 9° also indicates the significant angle-dependent field-induced phase transition.

Moreover, in the angle-field phase diagram of α -RuCl₃, the phase transition at $\theta \simeq 9^\circ$ is thought to be very close to the tricritical point between the zigzag AFM, quantum paramagnetic, and the QSL phases. Here,

the transition from zigzag order to the quantum paramagnetic state is labelled by H_c , and the transition from zigzag order to the QSL state is labelled by H_c^l . Near the tricritical point, the strongly competing tendencies may make the phase transition signal rather inconspicuous (*e.g.*, the featureless transition at the tricritical point in piezoelectric materials, show in Fig. R3). For α - RuCl_3 , two co-authors of this manuscript (H.L. and W.L.) recently reported a follow-up theoretical study suggesting that the transition signal in the Grüneisen parameter is also rather weak near the tricritical point (see Fig. R4). Specifically, the signals of order-disorder transition at H_c for $\theta \simeq 9^\circ$ and 20° are weaker than the those measured at other larger θ angles.

FIG. R3. As shown in the right panel, there exists a tricritical point at $x = 32.5$ in the piezoelectric material. For $x = 32.5$, we expect a transition signal at around 60°C , however, the peak signal observed is rather weak in the heat flow measurement shown in the left panel (EPL, **115**, 37001, 2016).

To provide further evidence supporting the data reliability, we followed the Reviewer's suggestion, as also suggested by Reviewer #3 in section #3-2, and measured the magnetization process for $\theta \simeq 90^\circ$ (*i.e.*, within the ab -plane) up to 90 T using the single-turn coil techniques. The results are shown in Fig. R5, which demonstrates that only the 7 T transition is present and no higher field transition is observed up to 90 T in-plane fields. Therefore, our measurements are reproducible as compared with the previous results (*e.g.*, by Kubota et al. in Ref. 27).

In summary: (1) For $\theta \simeq 0^\circ$, H_c^l and H_c^h transitions have been confirmed to be intrinsic property of ABC stacking sample based on the averaged dM/dH curve shown in Fig. R1; (2) For $\theta \simeq 9^\circ$, we determine more accurately the location of H_c based on the new experimental data obtained by the non-destructive magnet, consistent to previous results measured at 9° by Modic et al.; (3) The weak signals in the dM/dH curve at about 9° can be ascribed to the proximity to a tricritical point. (4) The in-plane measurements up to 90 T further evidence the data reliability. Therefore, based on our experimental measurements and model calculations, and with the previous results (by Modic et al. and Kubota et al.) reproduced, we humbly yet firmly believe that our conclusion is drawn on a solid ground. We believe that there exist two phase transitions for small tilting angles, with an intermediate phase possibly of QSL nature.

FIG. R4. The calculated Grüneisen parameters (Γ_B) reported in arXiv:2211.01360. From the behaviors in Γ_B one can determine the locations of B_{c1} and B_{c2} , and the two transitions merge at the tricritical point around 4° (with a relatively large error bar).

We have revised the manuscript accordingly to include the discussions above and enhance its readability.

FIG. R5. Magnetization process under magnetic field along ab -plane at 4.2 K, where only a single transition at around 7 T could be observed.

Reviewer #2-2 : Another point of concern is the H_c^{AB} transition, which the authors note is irrelevant. I think observing this transition suggests that a significant fraction of the sample has been damaged (gray

arrow in Figure S1), possibly during handling or preparing the samples, or by the large torques experienced in a magnetic field. I agree with the authors that when a large portion of the sample experiences ABAB stacking, the system becomes more isotropic and thus, one might expect the H_c^{AB} transition not to change much with angle. However, there are several concerning aspects about the observed anisotropy (or lack thereof) in other transitions. For example, can the authors explain why H_c^l in the near zero degree data is fairly isotropic in experiments i-iii? I presume H_c^l corresponds to the AFM transition observed by others, which we know is highly anisotropic. This weakens evidence that the H_c^l transition is the AFM transition—either the data is too noisy and switching noise is introducing additional features or the authors do not understand the origin of H_c^l . It does not help that the “irrelevant” H_c^{AB} transition (down-sweep data in Figure S4) is as large or larger than the H_c^l and H_c^h transitions that the authors claim is where readers should focus their attention.

Reply: Thank you very much for the comments. Indeed, as the Reviewer also agrees, we believe that an isotropic transition signal (which we denote as H_c^{AB}) would appear given ABAB stacking fault is present in the compound. Below we provide further evidence by measuring a sample full of ABAB stacking fault. We damage α -RuCl₃ by deforming the sample and producing lots of ABAB stacking fault components in the sample, and find the ordering temperature of the sample is about 14 K [see Fig. R6(a)]. Then we performed high-field experiments up to 100 T along the c^* axis on this sample and show the results in Fig. R6(b), where we find only H_c^{AB} peak at around 14(\pm 4) T.

This clearly resolves the concern on whether that H_c^l (or H_c^h) is caused by the transition of ABAB stacking fault component in the sample, as this ABAB stacking sample only has one phase transition at approximately 14 \pm 4 T, and the transition near 35 T does not appear in this sample. Therefore, we conclude that the H_c^l signal should belong to the ABC stacking component. This is also supported by the averaged dM/dH data (see Fig. R1), as we have discussed in the previous section.

Regarding the anisotropy of H_c^l , we would stress that the anisotropy (i.e., $\theta = 0 \pm 2.5^\circ$ angle dependence for experiments (i-iii)) of H_c^l is indeed found much weaker than that of H_c^h . Due to the strong electrical noise below 30 T, the value of H_c^l has an error bar of approximately 5 T in experiments, which is within a reasonable range of $\pm 2.5^\circ$ and is consistent with the model calculations. Matter of fact, when the field angle increases and goes across the tricritical point the highly anisotropic AFM transitions (reduced from about 35 T to 7 T) can be observed, which agrees with previous works.

We have added the new measurements (Fig. R6) and analysis in the revised manuscript, and hope this additional information further clarify our points and address the Reviewer’s concerns.

Reviewer #2-3 : *In summary, I cannot recommend this paper for publication as I think the experiments need to be better executed and the data is not convincing. I commend the authors on their efforts to obtain such a challenging dataset, but the lower inset in Figure 2 summarizes the fact that the authors cannot distinguish real transitions from the electrical noise in their measurement. I actually agree with the authors that it looks like there may be evidence for a transition near 80 tesla, but one should first confirm other*

FIG. R6. (a) The temperature-dependence in magnetization of the damaged α - RuCl_3 sample with lots of ABAB stacking components, whose properties are indeed strongly influenced. (b) The magnetization process of ABAB stacking sample under magnetic fields along the c^* axis at 4.2 K, where H_c^{AB} denotes the isotropic transition field in this sample.

known transitions at lower fields and thereby confirm that the 80 tesla transition is not the suppression of AFM order. I appreciate that other methods to study α - RuCl_3 in such high magnetic fields is limited, but without complementary information and with their current data as-is, it is not sufficient to claim that the intermediate field regime between H_c^l and H_c^h is a quantum spin liquid.

I hope that these measurements can be improved to reduce noise and substantiate the authors' claim in the future.

Reply: We understand the Reviewer's thoughts on the importance of more high-quality data to further support our claims. In the revised manuscript, we have provided additional evidence to support our conclusions, including the averaged dM/dH curves at 0° , the non-destructive measurements for 20° and 9° , in-plane magnetization process data for α - RuCl_3 up to 90 T, and the magnetization for the ABAB stacking sample under out-of-plane fields (i.e., along the c^* -axis). These results further support the existence of two

phase transitions at $\theta \simeq 0^\circ$. As the field angle increases, they come close to each other and eventually merge into one transition. Furthermore, weak signal in dM/dH curve at $\theta \simeq 9^\circ$ also suggests that the phase transition here is very close to the tricritical point between the AFM, quantum paramagnetic, and QSL states. The resemblance of the measured $H_c^{l(h)}$ to model calculations support our understanding of the zigzag-QSL and QSL-PM spin state transitions. Given these points, we hope that the Reviewer can be convinced and reconsider our manuscript for publication in Nature Communications.

Reviewer #2-4 : *Minor comments (some of these are repetitive to that written above, but sometimes it's useful to see it said in different words):*

- *Line 32, renowned honeycomb model “with bond-dependent exchange”*
- *Line 45, chlorine?*
- *I don't understand the last sentence of the caption for figure 2*
- *What is the reason for the downturn at the highest fields on the 9 deg data?*
- *Page 3, line 17-19 Why does H_c in the 9 deg data occur at higher field than H_c^l for 0 deg?*
- *Line 55, emergence*
- *Page 4, line 1-3, this summarizes the main problem with the paper because it suggests that data taken in a different magnet or at a different angle is qualitatively different, and one would expect a smooth evolution of these transitions. Furthermore, there is no overlap of consistency (or field range) between this data set and that found in the literature.*
 - *Page 4, Line5-10, just because it's disordered above H_c , doesn't mean it's a spin liquid. This is insufficient to make this claim.*
 - *Line 13; recent*
 - *In Figure 3, the H_c^l field seems angle independent in the data in the near zero curves while H_c^h changes a lot between experiments i-iii. How can you rule out that H_c^l is not related to ABAB stacking because even the calculation suggests that it should be very sensitive to angle. Also, in the calculation it follows the wrong angle dependence compared to the literature. Do you understand this?*
 - *Figure S4, where is the other gray curve? Why is only the derivative shown on the down sweep? The down sweep data looks less noisy so why isn't it used in the main text?*
 - *The main issue with Figure S4 is that the H_c^l transition is less pronounced than the $H_c^A B$ transition. The H_c^l transition is too near the noise in the up sweep data. For angles near the c^* axis, it's possible that the H_c^h transition is the H_c associated with AFM.*
 - *My apologies if I missed this in the manuscript, but what is the approximate size of the samples being measured?*

Reply: We thank the Reviewer for reading our manuscript carefully and giving us these constructive comments/suggestions. As there are 13 detailed comments in total, we list them as follows and address one by one.

#2-4.1 Line 32, renowned honeycomb model “with bond-dependent exchange”

Thanks, we now add the words “with bond-dependent exchange”.

#2-4.2 Line 45, chlorine?

Yes, thank you for pointing out the spelling mistake.

#2-4.3 I don’t understand the last sentence of the caption for figure 2

In the revised version, as we have provided more precise determination of H_c for data at 9° , we thus reorganize the description of the figure captions.

#2-4.4 What is the reason for the downturn at the highest fields on the 9 deg data?

We think it reflects the saturation of the magnetization as field increases. We add this remark in the revised manuscript.

#2-4.5 Page 3, line 17-19 Why does H_c in the 9 deg data occur at higher field than H_c^l for 0 deg?

Thank you for your question. In the previous manuscript, we roughly determined the highest kink in Fig. 2 inset as H_c with a big error bar ± 15 T. In the revised version, we define H_c based on the new measured data in Fig. R2, and give a more precise $H_c \simeq 25 \pm 5$ T with $\theta \simeq 9 \pm 2.5^\circ$.

#2-4.6 Line 55, emergence; #2-4.9 Line 13, recent

Thanks a lot for pointing out the typos.

#2-4.7 Page 4, line 1-3, this summarizes the main problem with the paper because it suggests that data taken in a different magnet or at a different angle is qualitatively different, and one would expect a smooth evolution of these transitions. Furthermore, there is no overlap of consistency (or field range) between this data set and that found in the literature.

In the main section of this reply, we have provided additional data to show the consistency between the experimental results measured by the single-turn coil magnet and the non-destructive magnet from 90° to 9° . Those results also show consistency with the previous reported results by Kubota et al. and Modic et al. as we discussed in #2-1 and #2-2.

We understand the concern of Reviewer “there is no overlap of consistency (or field range) between this data set and that found in the literature.” is that the phase transition H_c^l has not been observed at 35 T until now, except for Modic et al..

Here, to further address the concern of Reviewer, we also provide the dM/dH curves measured by non-destructive method at $\theta \simeq 0^\circ$ as shown in Fig. R7. In Fig. R7, there is a very sudden noise at 35 T in all of the three measurements. Based on our experience, this noise signal is very nontrivial which should not emerge at almost a same field strength in every measurements. This may be caused by a torque effect which has been reported by Modic et al., and the sudden torque effect makes the magnetization difficult to be measured by employing the non-destructive magnet.

However, by using the single-turn coil technique, this torque effect becomes insignificant as the pulse time is only $6 \mu\text{s}$ in the field generation, which help observe the H_c^l transition more clear in magnetization

measurements. With this we believe the discussions above can address the concern of the Reviewer.

FIG. R7. The dM/dH curves measured by the non-destructive magnet at 4.2 K for $\theta \simeq 0^\circ$. All three curves show a noise at 35 T, which is consistent with the H_c^l signal measured at 35 T under the single-turn coil magnetic field.

#2-4.8 Page 4, Line5-10, just because it's disordered above H_c , doesn't mean it's a spin liquid. This is insufficient to make this claim.

Thanks for the comment. We agree with the Reviewer here, yet firmly believe that the remarkable consistency between the present high-field experiments with model calculations constitutes a very strong indication that the proposal of intermediate QSL is realized in the compound α - RuCl_3 .

Having said that, we consider this comment very seriously and thus soften our claim here and revise the

title accordingly to “**Possible** Quantum Spin Liquid in the Kitaev Material α -RuCl₃ under High Magnetic Fields up to 100 T”.

#2-4.10 In Figure 3, the H_c^l field seems angle independent in the data in the near zero curves while H_c^h changes a lot between experiments i-iii. How can you rule out that H_c^l is not related to ABAB stacking because even the calculation suggests that it should be very sensitive to angle. Also, in the calculation it follows the wrong angle dependence compared to the literature. Do you understand this?

We note that the H_c^l is much less sensitive to the field direction in the small angle range than that for H_c^h , as seen in both experimental measurements and model calculations. As the θ errors are estimated to be $\pm 2.5^\circ$ based on the change of H_c^h , the difference of H_c^l in experiments (i-iii) should also be in the experimental error bar (± 5 T at around 30 T).

For the concern of ABAB stacking fault, as has been discussed in the response to **Reviewer #2-2** and shown in Fig. R6, we find a single peak at about 14 (± 4) T in the dM/dH curve of a ABAB stacking sample. Therefore, this H_c^l signal at around 35 T should be ascribed to the ABC stacking component.

For the latter comment, the calculation suggests that the high transition field H_c^h is very sensitive to the angle, but the lower one is not, consistent with the experiments. It is noteworthy that the model considered here in the 2D DMRG calculations is a minimum effective spin model, which only includes Kitaev K , off-diagonal Γ and Γ' , and nearest-neighbor Heisenberg J terms, and does an excellent job to capture the main feature in the compound. In realistic experiments, there may be also some other interactions like next-nearest-neighbor and third-neighbor Heisenberg couplings, as well as the inter-layer interactions that remains to be included in the model calculations. They may contribute to the subtle difference to previous experiments as the Reviewer concerned here, which certainly calls for future theoretical studies.

#2-4.11 Figure S4, where is the other gray curve? Why is only the derivative shown on the down sweep? The down sweep data looks less noisy so why isn't it used in the main text?

The reason that why we did not show the down-sweep integrated magnetization curves is a generally known technical issue in the community. In the down-sweep process, the dM/dH data is less noisy because it is far away from the electromagnetic switch noise in the time domain. However, during the down-sweep process, the single-turn coil has begun to deform due to Lorentz forces, causing the magnetic field to be inhomogeneous. This results in the down-sweep field waveform in two times are not as perfectly consistent as in the up-sweep field waveform, which may induce serious errors in the background measurement during the down-sweep process. In the case of large magnetization signals, the error of background caused by this inhomogeneity in the down-sweep process can be ignored. However, for the compound α -RuCl₃, the magnetization signal is relatively weak and the effect of this errors may become significant. Therefore, we can only trust the transition signals in the down-sweep dM/dH data, but not its integration for obtaining the magnetization. Therefore, we don't show the down-sweep data in the main text.

However, we show the down-sweep data in the supplementary materials, because the transition peak positions are reliable and useful for determining the transition fields.

#2-4.12 The main issue with Figure S4 is that the H_c^l transition is less pronounced than the H_c^{AB} transition. The H_c^l transition is too near the noise in the up sweep data. For angles near the c^* axis, it's possible that the H_c^h transition is the H_c associated with AFM.

Thank you very much for giving us opportunities to describe our data in different words. As we have explained in the comment #2-4.11, the down-sweep dM/dH curves suffer from background issue, which could not be used to draw the conclusion that H_c^l is “less pronounced than H_c^{AB} transition”. In fact, the height of H_c^l and H_c^{AB} are almost same based on the up-sweep data. If H_c^l transition is caused by the ABAB stacking, we should have observed a clear peak at ~ 35 T in Fig. R6(b) with similar height as the intrinsic peak (~ 10 T) in sample full of ABAB stacking fault; however, no signal appears there. Therefore, we believe H_c^l is intrinsic in α -RuCl₃. Additionally, as we have discussed, based on (1) the averaged dM/dH at 0° , (2) the almost featureless dM/dH curve at 9° , (3) the consistency between our lower phase boundary, the one reported by Modic et al. and Kubota et al., and the prediction of model calculations, we confirm there are two phase transitions at H_c^l and H_c^h , where H_c^l is associated with AFM order suppression and H_c^h relates to the polarization field.

#2-4.13 My apologies if I missed this in the manuscript, but what is the approximate size of the samples being measured?

In the single-turn coil experiments, the sample is cut to 0.9×0.9 mm² square. Several sample with ~ 0.2 mm thickness are stacking together to obtain enough thickness to measure the magnetization process. In the non-destructive pulse field experiment, the diameter of the sample is about 2 mm.

RESPONSE TO THE THIRD REVIEWER'S REPORT

Reviewer #3-summary : *In this manuscript Zhou et al. measured the magnetization of α -RuCl₃ under rotatable magnetic fields up to 95 T at 4.2 K. This wide field range is enabled by a specialized single-turn coil technique developed by the group over the past years. Compared with previous reports (e.g., Nat. Phys. 17, 240 (2021)), the major novelty of this manuscript is the observation of two magnetic transitions (H_c^l and H_c^h) under an out-of-plane field. The region of interest is in the middle of the two transitions ($H_c^l < H < H_c^h$), where DMRG simulations show that the zigzag magnetic order is suppressed by the field yet it is not strong enough to fully polarize the spins. This was taken as a hint of a possible quantum spin liquid phase. By rotating the field away from the out-field-plane direction, the two transitions seemingly merge into one, with the transition field shifting downwards in strength. Reassuringly, the magnetization of α -RuCl₃ measured in this manuscript under an in-plane field is similar to previous reports.*

Overall, I find the results quite interesting and the experiments carefully performed. The manuscript will surely add to the interest in a material system that has already received great attention. Therefore, I recommend the publication of the manuscript in Nature Communications, provided that the following

comments are properly addressed.

Reply: We thank the Reviewer for the precise summary, positive evaluation, and the recommendation of our work. Indeed, our results reveal the two-transition scenario that encloses an intriguing intermediate phase, which has been proposed to be a QSL phase in theory and now witnessed in high-field experiment. Such an intermediate phase has been intensively studied by applying in-plane fields, while our work opens a new avenue for finding such the exotic QSL state in α -RuCl₃. Below, we repose the comments and questions of the Reviewer one by one.

Reviewer #3-1 : *1. The data measured with field angles between 0° and 30° are not very convincing. Especially, the 9° and 20° magnetization curves have no obvious peaks in my opinion. I noticed that the 9° data was measured on a different sample with a different pickup coil. Can the authors provide more and higher quality data in this field range? I find it difficult to conclude that the two transition fields (H_c^l , H_c^h) at 0° merge into one transition at 9° from the data currently presented in Fig. 2, yet this is crucial for the interpretation of the data. In addition, when comparing to the DMRG calculations in Fig. 3, the experimental peaks at 9° and 20° are also much weaker than the theoretical counterparts.*

Reply: Thank you very much for the comments. We agree with the Reviewer that the signals H_c^l and H_c^h observed at 0°, 9° and 20° are weak, and thank the reviewer's attention to our efforts in measuring the dM/dH curves at 9°. Firstly, as the dM/dH curve at this angle was found to be featureless in experiments performed earlier by using the transitional pick-up coil, thus we measure it in a different pick-up coil (with a 1.4mm diameter) as presented in the manuscript. However, the newly developed pick-up coil has some serious technical problems which always gets broken in the single-turn coil experiment. Therefore, we apologize that we can not provide more and higher quality data by employing this 1.4-diameter pick-up coil.

Nevertheless, to address the concern of the Reviewer, we provide several other evidence to support that our conclusions are correct. We kindly refer the Reviewer for our responses in section #2-1 to address a similar concern from Reviewer #2. We show the averaged data from the experiment (i-iii) as shown in Fig. R1, which proves that the transitions at H_c^l and H_c^h are intrinsic in the sample. Fig. R6 indicates that these two phase transitions are from ABC stacking sample rather than the ABAB stacking fault sample. We also provide new experimental data measured by the non-destructive magnet in Fig. R2. Those experiments are performed for several times, and the averaged results make the H_c transitions at 9° and 20° more significant. Meanwhile, As the peak is also very broad, in the revised manuscript, we re-estimate the length of the error bar. We also discussed that the dM/dH curves with weak transition signals at $\sim 9^\circ$ indicate the proximity to a tricritical point around the angle. This is also consistent with the DMRG calculated results as shown in Fig. 3(b) and R4. Based on the additional information and discussion, we hope the conclusion of the intermediate QSL phase is more convincing to the respected Reviewer.

As for the comparison between the experiments and the calculations, the differences mentioned by the Reviewer on the height of peaks is understandable, as the DMRG method is performed on an effective

2D spin model with only intra-layer Kitaev interaction K , nearest-neighbor Heisenberg J , off-diagonal Γ and Γ' interactions taken into account. The effects of inter-layer couplings and related physics such as the stacking effect in the realistic α - RuCl_3 material are not included in the 2D model calculations. The reason for why the height of peaks obtained by the DMRG calculation is stronger than the experimental results might be caused by these possible terms/factors not considered in the present model, e.g., the next- and third-nearest neighbor Heisenberg couplings, the inter-layer interactions, and the inhomogeneous external field in the high-field measurements.

Reviewer #3-2 : 2. *Since most research interest in α - RuCl_3 is related to the magnetic behavior under an in-plane (rather than out-of-plane) field, it would certainly help the manuscript to show a high-in-plane-field magnetization curve using this special single-coil technique. This measurement is also important as a cross-reference to the out-of-plane curves.*

Reply: We thank the Reviewer for the very constructive suggestion, and have performed the magnetization measurement with the single-turn method. As shown in Fig. R5, we find there exists only one phase transition at around 7 T, without any other signals for further spin state transitions. We believe that these additional data serve as an important cross-check and include them in Fig. 1e in the revised manuscripts to make our main conclusion in the present work even more convincing.

Reviewer #3-3 : 3. *Can the authors provide a physical picture of why the two magnetic transitions appear as peaks in dM/dH curves? It is not very intuitive how to reconcile the magnetization slopes with the change in the zigzag/other magnetic orders.*

Reply: Thanks for the questions. The magnetic susceptibility $\chi = dM/dH$ in the low-field regime are small as the antiferromagnetic zigzag order competes against the uniform magnetic field that instead tends to align the spins. Meanwhile, under large fields, as the spins are polarized, the susceptibility also exhibits very small values in the paramagnetic states. Within the intermediate-field regime proposed to be a spin liquid states, the spins are disordered, which also has small susceptibility. In contrast, near the phase transition point the susceptibility χ diverge as they correspond to the second derivatives of energy and are expected to show singularities at the transition points. We hope our these explanations provide an intuitive understanding of the magnetization behaviors reflected in dM/dH .

Reviewer #3-4 : 4. *Other minor points:*

- *It is not easy to distinguish the M - H and dM/dH - H curves in Fig.1. Please use arrows to indicate the two sets of curves (similar to the fashion in Fig. S1).*
- *On page 2 of the SI, “note” should be “not”.*
- *On line 14, page 4 of the main text, “resent” should be “recently”. In addition, it is unclear what “high K -term” means.*
- *“(0, 30) T” in the last sentence of the caption to Fig. 3 is unclear. Please rephrase.*

Reply: Thanks a lot for drawing our attention to these problems. We have revised Fig. 1 and added

arrows to the two sets of curves, corrected the typos, changed the “The resent proposed high K-term microscopic spin model” to “The recently proposed microscopic spin model with large Kitaev coupling”, and rephrased “The (0, 30) T *M-H* data of destructive measurements are [...]” as “The *M-H* data of destructive measurements below 30 T fields are [...]”.

Overall, we thank the Reviewer #3 again for the very constructive comments/suggestions.

SUMMARY OF CHANGES

The summary of the major changes in the resubmitted text are shown below. Some spelling and grammatical modifications are not listed.

1. Title

We changed the title into “Possible Intermediate Quantum Spin Liquid Phase in the Kitaev Material α -RuCl₃ under High Magnetic Fields up to 100 T”.

2. Author

We added a new cooperator to the author list.

3. Affiliation

We updated some of Author affiliations.

4. Page-1: lines(20, 33, 34, 47, 57)

Some typos or unclear descriptions were corrected.

5. Page-2: Figure 1

We updated the Figure 1 and related figure captions.

6. Page-2: lines(10, 32); Page-3: lines(2-5, 22-26)

We reorganized the description related to the Figure 1.

7. Page-3: Figure 2

We update the Figure 2 and related figure captions.

8. Page-3: lines(38-46, 49-65, 66, 68); Page-4: lines(3, 7-12, 15-16, 19, 28-30, 32, 36-44, 46, 48-49, 51-58, 61-63, 65)

We reorganized the description related to the Figure 2.

9. Page-4: Figure 3

We updated the Figure 3 and related figure captions.

10. **Page-4: lines(76-77, 85-86); Page-5: lines(2, 12, 14-15, 17-28)**

Some typos or unclear description were corrected. We added some discussion for the comparison between experimental and calculated results.

11. **Page-5: Figure 4**

We updated the Figure 4.

12. **Page-5: line(33, 45-46, 59-60, 66, 81, 87-89)**

Some typos or unclear description were corrected. We also removed some unsuitable discussion.

13. **Page-6: line(15-19, 41-43)**

We added some description to show the size of sample measured in this work.

14. **Acknowledgements**

We thanked some of the people who helped in the follow-up experiment. The funding information was updated.

15. **Author contributions**

The author contributions were updated based on the revised version.

16. **Supplemental Materials**

Some details are updated.

REVIEWER COMMENTS

Reviewer #1 (Remarks to the Author):

Authors have addressed the revisions satisfactorily, and I recommend proceeding with the publication.

Reviewer #2 (Remarks to the Author):

Thank you to the authors for the explanations, and for addressing many concerns. In particular, I appreciate that all changes to the text were made in blue color. This is very useful. I also appreciate the “summary of changes” included at the end of the report.

Unfortunately, I am still not very convinced by the data. The new data at 9 and 20 degrees shows a transition at 10 tesla that is as big or bigger than the higher field transitions that the authors want to convince is related to ABC stacked RuCl₃. If the 10 tesla feature is really related to ABAB stacking, then this means that a significant part of the sample is not ABC stacked.

In addition, the averaging of the three measurements done at zero degrees does not really convince me that there are two transitions observed. The noise just below 35 tesla in figure 1 in all gray curves is a great concern; it's magnitude is much greater than the 35 tesla transition. The fact that the lower field transition as illustrated in Figure 1d looks like a peak is really only dependent upon how the authors choose to draw the “guide to the eye”. Without the dashed line, one would likely not say that a peak exists.

To summarize, I am still concerned about the crystal quality due to ABAB stacking and unfortunately, I think the data is too noisy to be convincing. I am, however, very pleased to see that the authors have toned down their claim that the intermediate state is a QSL and I am impressed by the efforts made for the additional data. If this paper were accepted, I would still like to see some discussion related to the possibility that the feature observed at ~80 tesla could be the suppression of AFM order. I find the appearance of the 80 tesla transition more convincing than the 35 tesla transition, which is why I wonder if there really are two transitions in the 0 degree data. Perhaps one way to further convince readers of both the presence and nature of these transitions would be to increase the temperature above the AFM transition to see how they compare with the lower temperature

data. Certainly if it's related to AFM, it should disappear at higher temperatures, but I don't know what one should expect for the purported QSL.

Reviewer #3 (Remarks to the Author):

The authors have satisfactorily addressed my questions. I recommend the publication of the manuscript in Nature Communications.

RESPONSE TO THE FIRST AND THIRD REVIEWER'S REPORT

Reviewer #1: Authors have addressed the revisions satisfactorily, and I recommend proceeding with the publication.

Reviewer #3: The authors have satisfactorily addressed my questions. I recommend the publication of the manuscript in Nature Communications.

Reply: We thank for the recommend of the publication of Reviewer #1 and #3, and also thank for their appreciation for our work.

RESPONSE TO THE SECOND REVIEWER'S REPORT

Reviewer #2-1: Thank you to the authors for the explanations, and for addressing many concerns. In particular, I appreciate that all changes to the text were made in blue color. This is very useful. I also appreciate the “summary of changes” included at the end of the report.

Reply: We are glad that the reviewer appreciates our efforts on the previous reply.

Reviewer #2-2: Unfortunately, I am still not very convinced by the data. The new data at 9 and 20 degrees shows a transition at 10 tesla that is as big or bigger than the higher field transitions that the authors want to convince is related to ABC stacked RuCl₃. If the 10 tesla feature is really related to ABAB stacking, then this means that a significant part of the sample is not ABC stacked.

Reply: We appreciate the Reviewer's comments regarding the ABAB feature at 10 Tesla, and we understand his or her concerns. However, we believe the inevitably existing ABAB stacking in the sample does not change our conclusion here. In Fig. R1, we present further evidence showing that the higher field transition at H_c^l indeed reflects the intrinsic magnetic properties of ABC stacking, and compared the results to the measurements on a sample full of ABAB stacking. We hope the reviewer could be convinced with these additional experimental results.

In Figure R1, the blue curve represents the averaged dM/dH curve measured at 9 degrees, which is also shown in Fig. 2 in the manuscript. We obtained the red curve at 9 degrees from a sample predominantly composed of ABAB stacking components, using the same experimental setup as the blue curve.

By comparing the blue and red curves, we conclude that the high field transition around 28 T is caused by the ABC stacking component, even though the dM/dH curve still exhibits a significant ABAB stacking feature.

Figure R1 The dM/dH curve obtained at 9° for two samples: blue one has both ABC and ABAB stacking components and the other with predominantly ABAB stacking components. Both curves have been normalized using the peak observed at 10 Tesla.

Reviewer #2-3: In addition, the averaging of the three measurements done at zero degrees does not really convince me that there are two transitions observed. The noise just below 35 tesla in figure 1 in all gray curves is a great concern; it's magnitude is much greater than the 35 tesla transition. The fact that the lower field transition as illustrated in Figure 1d looks like a peak is really only dependent upon how the authors choose to draw the "guide to the eye". Without the dashed line, one would likely not say that a peak exists.

Reply: We appreciate the comments and discussion provided by the Reviewer. We believe that the concern raised by the Reviewer in this regard is directly related to his/her previous comment. As we have demonstrated in Fig. R1, it is rather natural to associate the H_c^l transition at 9 degrees in the vicinity of 35 T with the ABC stacking component. However, we understand the Reviewer's concern as the signal observed around 35 T is weak.

To address the Reviewer's concern more comprehensively, we have conducted additional magnetization process measurements at 0 degrees up to 100 T. We assume that the Reviewer is familiar with pulse magnetic field experiments, and thus we provide more detailed information about the new experiment to further clarify the H_c^l transition at 0 degree.

Firstly, we present in Fig. R2 the time dependence of the magnetic field and the corresponding dM/dt curves. The methodology for obtaining the data in Fig. R2 is described in Ref. [Physical review letters 111 (13), 137204]. In the up-sweep of the magnetic field, three features are observed in the dM/dt curve (blue curve), i.e., a peak labeled as H_c^{AB} , a shoulder labeled as H_c^l , and the peak labeled as H_c^h (later, we will explain H_c^h is not a peak with respect to the magnetic field). Those three features are also observed at the down-sweep process of the field (green curve), indicating three possible transitions observed up to 100 T. We should note that the shoulder at H_c^l in the up-sweep

process is not very clear because of the starting noise problem.

Figure R2 Time dependence of the magnetic field and the dM/dt curves observed at 0-degree field direction.

To address the concern from the Reviewer about the existence of transition field H_c^l at 0 degree, we present the dM/dH curve for the down-sweep process in Figure R3. In the figure, we can still observe the presence of the H_c^{AB} feature. However, considering the previous discussion and the results shown in Figure R1, we can conclude that this ABAB component does not contribute any additional signals in the high field region. The H_c^l transition is clearly observed in this dM/dH curve, which we believe should be ascribed to the ABC stacking components.

The phase transition of H_c^h has not been fully observed, i.e., only the starting feature of the transition peak is observed. Sometimes this kind of behavior can be attributed to a divergence caused by a technical issue, namely, due to the vanishing denominator dH/dt in $dM/dH = (dM/dt)/(dH/dt)$, where dH/dt becomes zero at the maximum point of the magnetic field. To rule out this possibility, we present both the up-sweep and down-sweep dM/dH curves in the inset. The overlap of the dM/dH curves between the up-sweep and down-sweep clearly indicates that the upward behavior around 95 T is not due to the technical issue, but rather reflects an intrinsic property of the H_c^h transition.

We find that both the signals at H_c^h and H_c^l transitions in the newly performed experiment are larger than the previously measured results, and attribute this to the strong magnetic anisotropy associated with these two-phase transitions and the improved alignment between the magnetic field and the c^* axis of the sample in the new setup. More specifically, the present experiment seemed to be performed at a nearly ideal 0 degree although it happened by chance. This result also supports the theoretical predictions on the angle-field phase diagram in our manuscript. The H_c^h peak is expected to be located at higher than 100 T in model calculations, and the dot-dashed curve is a guide to the eye as reflecting this fact.

We believe that the newly provided data adequately addresses the Reviewer's concerns regarding the existence of the H_c^l transition. Furthermore, if this manuscript can be accepted for publish, we

would like to make these results openly available in the peer review materials.

Figure R3 The dM/dH curve at 0-degree. The gray dashed curve is just a guide to the eye. The inset show the comparison of dM/dH curves between the up-sweep and down-sweep.

Reviewer #2-summary: To summarize, I am still concerned about the crystal quality due to ABAB stacking and unfortunately, I think the data is too noisy to be convincing. I am, however, very pleased to see that the authors have toned down their claim that the intermediate state is a QSL and I am impressed by the efforts made for the additional data. If this paper were accepted, I would still like to see some discussion related to the possibility that the feature observed at ~80 tesla could be the suppression of AFM order. I find the appearance of the 80 tesla transition more convincing than the 35 tesla transition, which is why I wonder if there really are two transitions in the 0 degree data. Perhaps one way to further convince readers of both the presence and nature of these transitions would be to increase the temperature above the AFM transition to see how they compare with the lower temperature data. Certainly, if it's related to AFM, it should disappear at higher temperatures, but I don't know what one should expect for the purported QSL.

Reply: We are pleased with the Reviewer's appreciation of our efforts in the previous reply. Based on the newly provided experimental results in this reply, we believe the main concerns raised by the Reviewer regarding the ABAB stacking and data quality have been addressed.

Furthermore, we acknowledge that the possibility that the 80 T phase transition may suppress the antiferromagnetic (AFM) order raised by the reviewer cannot be fully ruled out in the present work. However, our experimental and computational results, combined with those previously reported data by Modic et al, provide strong evidence that the H_c^l transition is an intrinsic characteristic of the ABC stacking component, and more likely represents a suppression of the AFM order. We have added related discussions in the revised manuscript, and thanks for the constructive suggestion.

Additionally, we acknowledge the Reviewer's comment regarding the potentially large error bar

associated with the 35 T transition at 0 degree due to strong magnetic anisotropy. We have confirmed this through our newly conducted experiment: It is worth noticing that the presence of a special strong magnetic anisotropic phase diagram consistently leads to a larger fluctuation of the experimental result in the vicinity of 0 degree.

Regarding the Reviewer's mention of temperature-dependent experiments, we believe that the QSL phase is also suppressed at higher temperatures and it would be hard to distinguish the role of the AFM and QSL with higher temperature experiments. We apologize for the unavailability of temperature-dependent measurements in this study because the experiment requires considerable time and efforts.

Based on the valuable suggestions and the comprehensive discussion above, we have reorganized the discussion section of the manuscript. We hope that the Reviewer could be convinced and will consider recommending the publication of our manuscript.

SUMMARY OF CHANGES

The summary of the major changes in the resubmitted text are shown below.

1. Page-4 Lines (62-71)

We added some discussions about the scenario that H_c^h corresponds to the transition field suppressing the AFM order.

2. Page-5 Figure 4

Two data points were attached based on the experimental results in this response.

3. Page-5 Figure 8 (Acknowledgements)

The funding information was updated.

REVIEWERS' COMMENTS

Reviewer #2 (Remarks to the Author):

Thank you for the detailed reply, and the time taken to produce more data.

I find the data and explanations surrounding Figures R2 and R3 much more convincing. I understand the concern regarding the possibility of the high field upturn being related to the experimental timescales, and the reason to include the downsweep to confirm that it is indeed intrinsic. I agree that it does look like the onset of a very high field transition. Similarly, I agree that the H_c^h transitions plotted in Figure 1 in all previous versions of the manuscript, as well as this one, also look convincing. It is also nice to see that the onset occurs at higher fields for the best aligned sample compared to the previous measurements, as expected from theory.

I am still not fully convinced about the stacking arguments put forth, but I appreciate that sample quality in RuCl₃ is a difficult component of this work. I don't believe that the community has a good grasp of how to relate the sample properties/transitions to the inherent stacking discrepancies. Importantly, this study shows that there is a feature appearing between 80-120 T in RuCl₃ and it should provide incentive to continue experimental work, both to improve the sample quality and make more measurements to better understand RuCl₃ in high fields.

I agree with the authors that the more recent data should be published, either in the peer review materials or in the supplementary. I recommend the paper for publication.